# Prevalence of Venous Thromboembolism in Intensive Care Units: A Meta-Analysis

**DOI:** 10.3390/jcm11226691

**Published:** 2022-11-11

**Authors:** Xiaoyu Gao, Liangnan Zeng, Haorun Wang, Shan Zeng, Junjie Tian, Ligang Chen, Tangming Peng

**Affiliations:** 1Graduate School, Southwest Medical University, Luzhou 646000, China; 2Department of Neurosurgery, Affiliated Hospital of Southwest Medical University, Luzhou 646000, China; 3Department of Nursing, The Fifth Affiliated People’s Hospital of Chengdu University of Traditional Chinese Medicine, Chengdu 610000, China

**Keywords:** venous thromboembolism, intensive care unit, meta-analysis

## Abstract

Objective: Venous thromboembolism (VTE) is a life threating complication in intensive care units (ICUs). This study aimed to pool the prevalence of VTE and examined the risk factors of VTE in intensive care patients worldwide. Methods: A systematic search in PubMed, EMBASE and Web of Science databases was performed. Studies reported that the data on the prevalence of VTE or relevant information were synthesized using a random-effects model. Results: A total of 42 studies reporting on 27,344 patients were included. The pooled prevalence of VTE was 10.0% (95% CI: 7.0–14.0%). Subgroup and metaregression analyses found that thromboprophylaxis strategy, simplified acute physiology score (SAPS II), age, study quality, sample size, malignancy, sex, spinal cord injury and injury severity score (ISS) moderated the prevalence of VTE in intensive care patients. Conclusions: The present meta-analysis revealed a high prevalence of VTE in critically ill patients. The risk factors of VTE included thromboprophylaxis strategy, SAPS II, age, malignancy, sex, spinal cord injury and ISS. Therefore, we need to pay more attention to high-risk populations of VTE in intensive care patients.

## 1. Introduction

Venous thromboembolism (VTE) driven by dysregulated coagulation, including deep-vein thrombosis (DVT) of the leg or pelvis and pulmonary embolism (PE), is a life-threating complication leading to morbidity and mortality all over the world, especially in critically ill patients. Similarly, it has also remained one of the most common unsuspected diagnoses found at autopsy in intensive care unit patients [1,2,3].

Afflicting approximately 10 million persons worldwide in all countries every year, VTE was a substantial contributor to the global burden of disease [4]. The prevalence of VTE in ICU patients ranged from 0.4% to 82.3% worldwide, which was the highest in Canada (400 in 100,000 persons) and the lowest in Thailand (82,300 in 100,000 persons) [5,6]. Baylis et al. indicated that VTE increased steadily over the last two decades [7]. In addition, Nobre et al. showed that 30% of patients had a recurrence of VTE in the last 10 years [8].

Previous studies have demonstrated that many factors could increase the risk of VTE in critically ill patients. Gregson et al. showed that older age, smoking and adiposity were consistently associated with higher VTE risk [9], and Krenitsky et al. suggested that factors with the highest VTE risk included transfusion and infection [10]. Furthermore, data showed that other factors, including trauma, major surgeries, malignancy, the inability of free movement, abnormal coagulation and indwelling venous catheters, were related with the increasing risk of VTE [11,12,13,14,15]. VTE is a fatal complication that, once occurred, can finally lead to a rise in mortality and disability, increase substantial healthcare costs and even sudden death [16,17]. Lutsey et al. showed that 3.5% (5943 in 170,021) developed pulmonary hypertension (PH) in the 2 years after their first VTE incident [18]. Makedonov et al. indicated that 20 to 50% of patients developed post-thrombotic syndrome (PTS) after a proximal VTE, and patients with critical PTS can have a quality of life similar to those with malignancy, angina, and congestive heart failure [19].

VTE has been documented among critically ill patients in various studies; however, it was likely due to the disparities in screening methods, among other study-specific features, which left VTE’s accurate prevalence unknown. Therefore, this study aimed to pool the prevalence of VTE and examine the risk factors of VTE in ICU patients.

## 2. Methods

### 2.1. Literature Search

This meta-analysis was conducted by the Preferred Reporting Items for Systematic Reviews and Meta-Analyses (PRISMA) guidelines [20]. Two investigators (XYG and LNZ) independently and systematically searched the databases, including PubMed, EMBASE and Web of Science, from their inception to 26 September 2022, using the following search terms: “venous thrombosis” OR “venous thromboembolic” OR “venous thromboembolism” OR VTE OR “deep venous thrombosis” OR DVT OR “lung embolism” OR “pulmonary embolism” OR PE AND “critical care” OR “critically ill” OR “intensive care” OR ICU AND “rate OR prevalence OR percentage”. We registered our protocol on PROSPERO accessed on 16 February 2022. (https://www.crd.york.ac.uk/PROSPERO/#index.php).

### 2.2. Study Selection and Data Extraction

Original studies were included if they met the following criteria: (1) observational studies including cross-sectional, case-control or cohort studies (only the baseline data were extracted for analyses in the last) conducted among ICU patents; (2) reported the data on prevalence of venous thromboembolism or relevant information that could generate the prevalence of venous thromboembolism in ICU departments; (3) being published in English; (4) studies in a special period (Corona Virus Disease 2019 and Severe Acute Respiratory Syndrome) or on special populations (the infants and adolescents) were excluded. The study selection and data extraction were performed by the same two investigators (XYG and LNZ) independently. Any disagreements were consulted by a third author (TMP). The first step was to screen the titles and abstracts independently to identify the relevant articles. Then, the full texts were reviewed for eligibility. If more than one study was published on the same data set, the one with more information or the larger sample size was included for analysis. A standardized Excel sheet was used to extract the following demographic and clinical characteristics: the first author, year of publication, country, year of survey, sampling method, department, sample size, age, male, body mass index (BMI), obesity, malignancy, abnormal coagulation, lower extremity fracture, pelvic fracture, spinal cord injury, thromboprophylaxis strategy, operations, central venous catheter (CVC), intensive care unit (ICU) length of stay (LOS), acute physiology and chronic health evaluation (APACHE II score), injury severity score (ISS), simplified acute physiology score (SAPS II) and quality score.

### 2.3. Quality Evaluation

The study quality was assessed by two independent investigators (XYG and LNZ) using the instrument for evaluating the methodological quality of epidemiological studies [21] with the following items: sampling methods; response rate; the definition and representative of targeted population; definition of VTE and validation of assessment instrument.

### 2.4. Statistical Analyses

The Comprehensive Meta-Analysis statistical software version 2.0 was used for the statistical analyses. The random-effect model was used to pool the prevalence of VTE [22]. Heterogeneity across studies was measured with *I*^2^ statistic (a significant heterogeneity was considered when *I*^2^ > 50%) [23]. Subgroup analyses and metaregression were conducted to explore the possible sources of heterogeneity. The following categorical variables were analyzed in subgroup analyses, in which the numeric categorical variables were categorized by the median splitting method: (1) year of survey: before 2013 vs. after 2013; (2) geographic region: Asia vs. European vs. North America vs. Oceania; (3) percentage of obesity: >26.8 vs. ≤26.8; (4) percentage of abnormal coagulation: >39.18 vs. ≤39.18; (5) percentage of lower extremity fracture: >23.04 vs. ≤23.04; (6) percentage of pelvic fracture: >20.10 vs. ≤20.10; (7) percentage of operations: >28.74 vs. ≤28.74; (8) percentage of CVC: >54.29 vs. ≤54.29; (9) ICU LOS: <8 vs. ≥8; (10) type of thromboprophylaxis strategy: pharmacological prophylaxis vs. mechanical prophylaxis vs. both; (11) SAPS II: >40.4 vs. ≤40.4. Continuous variables, including age, study quality, sample size, malignancy, male, spinal cord injury and ISS, were analyzed by meta regression analysis. Funnel plot and Egger’s regression model [24] was used to assess publication bias. Sensitivity analyses were performed by excluding one study in each turn to test the robustness of the overall pooled estimates. Statistical significance was defined as *p* < 0.05 (two-tailed).

## 3. Results

### 3.1. Characteristics of Included Studies and Quality Assessment

The details of the literature search and study selection is shown in Figure 1. Initially, 6726 papers were identified. After reviewing titles and abstracts, the full texts of 297 papers were read for eligibility. Finally, 42 studies covering 27,344 individuals were included in the meta-analysis (Table 1). Among these, 22 studies enrolled 8837 patients conducted in ICUs and 20 studies enrolled 18,507 patients conducted in specialist ICUs (including two in CICUs, seven in SICUs, two in TICUs, two in NICUs, five in MICUs, one in EICUs and one in LICUs). The percentage of males ranged from 25.45% to 100% and the mean age ranged from 31 to 64 years.

The quality assessment revealed that the mean score was 4.70 (ranging from 3 to 6). The targeted population was defined clearly in all articles, and the definition of VTE in 8 studies was not defined, while 24 studies did not provide sampling methods and 19 studies had no response rate or the response rate was lower than 70%. The study population in all studies was representative.

### 3.2. Prevalence of VTE in ICU Patients

The pooled prevalence of VTE in ICU patients from 42 studies while using the random-effect model was 10.0% (95% CI: 7.0–14.0%, *I*^2^ was 97.84%; Figure 2).

### 3.3. Subgroup Analyses and Meta-Regression

Results of subgroup analyses are shown in Table 2. Thromboprophylaxis strategy and SAPS II were significantly associated with the prevalence of VTE. The prevalence of VTE was highest in mechanical prophylaxis (31.9%, 95% CI: 25.8–38.6%) followed by used simultaneously (13.4%, 95% CI: 7.6–22.7%) and pharmacological prophylaxis (9.3%, 95% CI: 6.0–14.1%), while the VTE prevalence in ICU patients with mean SAPS II ≤ 40.4 (29.6%, 95% CI: 24.0–36.0%) was higher than that in those with SAPS II > 40.4 (13.0%, 95% CI: 8.8–18.8%). Year of survey, country, obesity, abnormal coagulation, lower extremity fracture, pelvic fracture, operations, CVC and ICU LOS were not associated with the prevalence of VTE (all *p* values > 0.05).

Meta regression analyses revealed that age (B = 0.04455, z = 11.99, *p* < 0.001), study quality (B = −0.09251, z = −3.36, *p* < 0.001), sample size (B = −0.00030, z = −22.41, *p* < 0.001), malignancy (B = −0.01705, z = −8.23, *p* < 0.001), sex (B = −0.00984, z = −4.08, *p* < 0.001), spinal cord injury (B = 0.07058, z = 7.45, *p* < 0.001) and ISS (B = 0.16487, z = 12.99, *p* < 0.001) were identified as significant moderators which contributed to heterogeneity cross-studies (Appendix A Appendix A).

### 3.4. Publication Bias and Sensitivity Analysis

Egger’s test (t = 0.60, 95% CI: −6.55–3.56%, *p* = 0.55) and funnel plot (Figure 3) did not detect publication bias. Sensitivity analyses showed no outlying study could change the robustness of results (Appendix A Appendix A).

## 4. Discussion

VTE has been documented among ICU patients in various studies. However, the reported prevalence of VTE in ICUs varied wildly, and the accurate prevalence of VTE remained unknown. In this meta-analysis, the overall prevalence of VTE among 27,344 individuals in ICUs was 10.0% (95% CI: 7.0–14.0%), ranging from 0.4% to 82.3%. This result showed a high rate of VTE in critically ill patients, which increases the mortality. The prevalence of VTE among ICU patients varied substantially, with the highest rate of 82.3% in SICUs [38]. It indicated that patients with surgical disease were more likely to develop thrombosis. Apart from the impact of different departments, discrepancy in diagnosis methods and socio-demographic characteristics could partly account for the differences between studies.

In this meta-analysis, the prevalence of VTE was moderated by different thromboprophylaxis strategy and SAPS II. Thromboprophylaxis strategies to prevent VTE were common in clinical work. However, the best strategy to prevent VTE remained controversial. In this study, pharmacological prophylaxis was associated with lower prevalence of VTE, which might show that pharmacological prophylaxis was more effective than mechanical prophylaxis or use in both to prevent VTE. The results were similar to the existing research, which indicated that anticoagulants can effectively prevent blood hypercoagulability via the inhibition of thrombin and coagulation factors, and consequently prevent VTE [65]. As also found in previous studies, the SAPS II was associated with disease severity, and the higher the SAPS II score, the more serious the disease [66]. However, in our study, the SAPS II ≤ 40.4 was associated with higher prevalence of VTE. This was partly because few studies were included in these subgroup analyses, and this finding needs to be re-examined in future studies.

Meta-regression analyses found that studies with a larger sample size (B = −0.00030, z = −22.41) were associated with a lower prevalence of VTE in this study. There was no explanation for this result except that the findings of the studies with a larger sample size might be more stable. In this study, malignancy (B = −0.01705, z = −8.23) was associated with a lower prevalence of VTE. This finding seemed unexpected, since malignancy was reported as a risk factor of VTE. However, it might be attributed to patients with malignancy often receiving more attention from medical staff, and they may also have received more preventive treatments [67]. In addition, studies with a higher percentage of males (B = −0.00984, z = −4.08) had a lower prevalence of VTE. Unlike males, this was probably because females often experienced pregnancy and puerperium, oral contraception and hormone therapy [15], which put female blood into a hypercoagulable state, thereby becoming more prone to develop VTE when ill.

This study found that older age (B = 0.04455, z = 11.99) was associated with a higher prevalence of VTE. It indicated that patients with older age were more likely to develop chronic disease and disability, which leads to hypercoagulability and limited bodily movement [68,69,70,71] when suffering from diseases such as trauma, major surgeries and malignancy, and which increased the prevalence of VTE. Thus, we should pay more attention to this special group. The prevalence of VTE was higher in patients with spinal cord injuries (B = 0.07058, z = 7.45). Stasis and hypercoagulability were two major risk factors for spinal-cord-injured patients [72]. In addition, spinal-cord-injured patients often needed a prolonged bed stay to reach neurological and functional recovery. All these factors might eventually lead to them developing thrombosis [73]. This study also showed that high ISS (B = 0.16487, z = 12.99) was associated with a higher prevalence of VTE, owing to the fact that a higher ISS often indicated more severe injury, which might increase the prevalence of VTE [74].

VTE was a fatal complication of patients with trauma, major surgeries, malignancy, the inability to free movement, and the like. Once VTE happened, it would increase the mortality and disability, increase substantial healthcare costs and even sudden death. Therefore, it was vitally importance to prevent and treat VTE in clinical work. Helms et al. [75] indicated that mechanical and pharmacological prophylaxis was suggested for critically ill patients. Current mechanical strategies include graduated compression stockings (GCS) and active devices such as intermittent pneumatic compression (IPC) [76,77,78]. With regard to the type of pharmacological thromboprophylaxis, low-molecular-weight heparin (LMWH) might have superior efficacy compared to unfractionated heparin (UFH) [79]. However, a high bleeding risk was considered a contraindication to anticoagulation. It was necessary to developed safety thromboprophylaxis strategies to prevent VTE.

There were several limitations to this meta-analysis. First, some of the sample sizes of the included studies were generally small, and most of the studies were retrospective studies. Second, the thromboprophylaxis strategy of VTE was different across the studies, which might influence the pooled prevalence of VTE. However, subgroup analyses could solve this confounding effect. Finally, heterogeneity in subgroup analyses remained, as it was unavoidable in meta-analysis of epidemiological surveys [80,81].

In conclusion, the prevalence of VTE was particularly high, especially in surgical disease patients. VTE was common in intensive care units worldwide and moderated by certain demographics and clinical factors, including age, sex, thromboprophylaxis strategy, SAPS II, malignancy, spinal cord injury and ISS. Considering the negative impacts of VTE on diagnosis, treatment, mortality and prognosis, we need to pay more attention to the high-risk populations of VTE. Methods to prevent VTE are now pharmacological and/or mechanical, such as GCS, IPC and LMWH, and they require the right time, dosage and duration, while also needing to be accompanied by good clinical care. However, more effective, detailed and individualized measures, such as an early assessment scale and specialized rehabilitation, should be taken to reduce VTE in intensive care units.

## Figures and Tables

**Figure 1 jcm-11-06691-f001:**
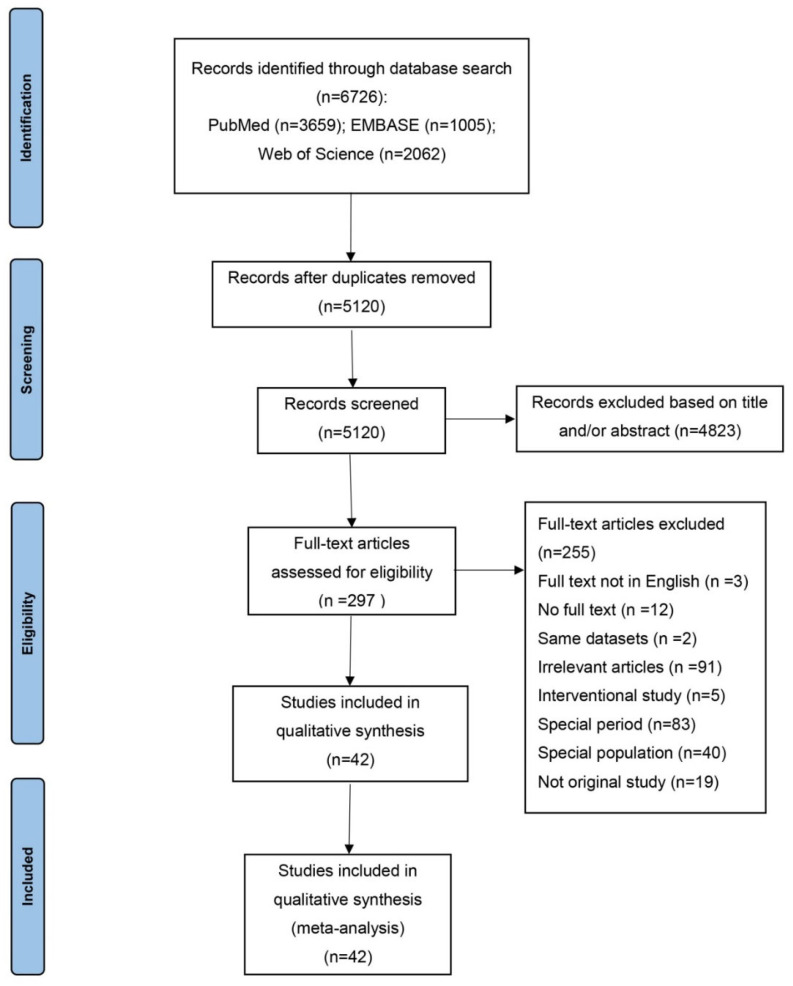
Flowchart of study selection.

**Figure 2 jcm-11-06691-f002:**
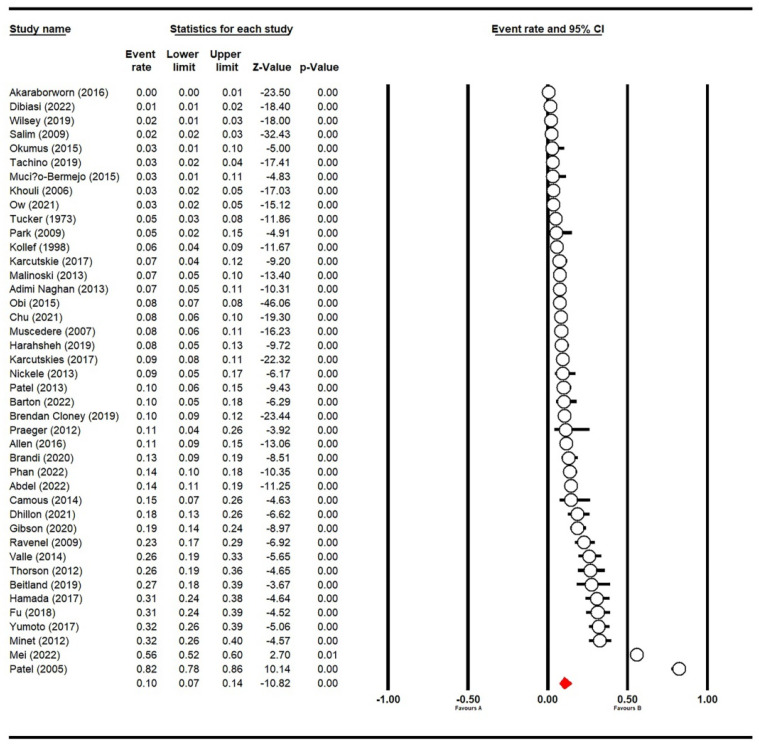
Forest plot of prevalence of VTE in ICU patients (*n* = 42) [5,6,25,26,27,28,29,30,31,32,33,34,35,36,37,38,39,40,41,42,43,44,45,46,47,48,49,50,51,52,53,54,55,56,57,58,59,60,61,62,63,64]. The red symbol indicated the prevalence of VTE in ICU patients.

**Figure 3 jcm-11-06691-f003:**
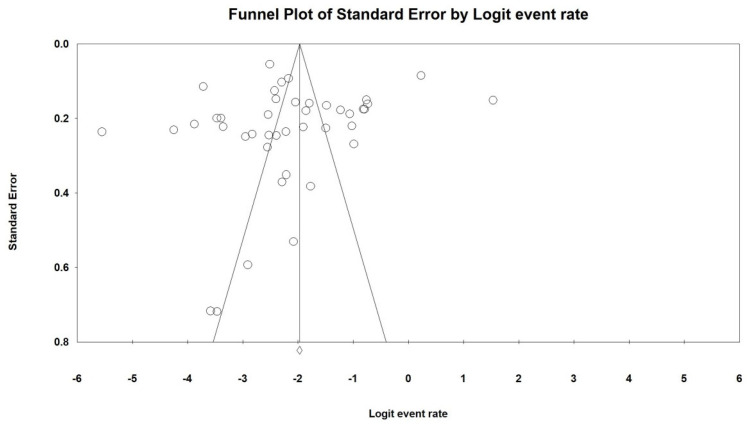
Funnel plot of publication bias on prevalence of VTE in ICU patients.

**Table 1 jcm-11-06691-t001:** Characteristics of studies included in this meta-analysis.

No.	First Author (Year)	Reference	Country	Year of Study Conducted	Sampling Method	Department	Sample Size	Male	Mean Age	BMI	Thromboprophylaxis Strategy	SAPS II Score	Malignancy	Spinal Cord Injury	ISS	Quality Score
1	Tucker (1973)	(Tucker et al., 1973) [25]	Britain	1970–1971	NR	CICU	342	342	56.5	NR	Pharmacological prophylaxis	NR	NR	NR	NR	3
2	Kollef (1998)	(Kollef et al., 1998) [26]	USA	1995	NR	MICU	323	164	57.5	NR	NR	NR	NR	NR	NR	4
3	Patel (2005)	(Patel et al., 2005) [5]	Canada	2000	consecutive	ICU	299	159	63.1	NR	Both *	NR	NR	NR	NR	5
4	Khouli (2006)	(Khouli et al., 2006) [27]	USA	1999–2001	consecutive	MICU	801	NR	NR	NR	NR	NR	NR	NR	NR	3
5	Muscedere (2007)	(Muscedere et al., 2007) [28]	Canada	2001–2002	consecutive	ICU	600	362	59.5	NR	Both *	NR	NR	NR	NR	6
6	Salim (2009)	(Salim et al., 2009) [29]	USA	1998–2005	NR	SICU	3286	2609	36.7	26.9	NR	NR	NR	NR	19.3	2
7	Ravenel (2009)	(Ravenel et al., 2009) [30]	USA	2004–2006	consecutive	ICU	181	107	54.3	NR	Pharmacological prophylaxis	NR	NR	NR	NR	6
8	Park (2009)	(Park et al., 2009) [31]	USA	2004–2005	NR	ICU	58	44	48.6	NR	Both *	NR	NR	NR	20.2	4
9	Thorson (2012)	(Thorson et al., 2012) [32]	USA	2009–2012	NR	TICU	106	78	47.0	NR	Both *	NR	2	5	30.0	3
10	Praeger (2012)	(Praeger et al., 2012) [33]	Australia	2008	NR	ICU	36	28	40.3	NR	Both *	NR	NR	NR	NR	5
11	Minet (2012)	(Minet et al., 2012) [34]	France	2009–2011	consecutive	ICU	176	109	NR	NR	Pharmacological prophylaxis	NR	16	NR	NR	6
12	Adimi Naghan (2013)	(Adimi Naghan et al., 2013) [35]	Iran	2006–2008	consecutive	MICU	243	141	NR	NR	Pharmacological prophylaxis	NR	NR	NR	NR	5
13	Malinoski (2013)	(Malinoski et al., 2013) [36]	USA	2008–2009	NR	SICU	411	296	48.0	28.1	Both *	NR	13	18	22.0	3
14	Nickele (2013)	(Nickele et al., 2013) [37]	USA	2009	consecutive	NICU	87	64	54.1	NR	Both *	NR	NR	NR	NR	4
15	Patel (2013)	(Patel et al., 2013) [38]	USA	2008–2009	NR	SICU	204	116	59.0	27.0	Both *	NR	54	NR	NR	3
16	Camous (2014)	(Camous et al., 2014) [39]	France	2003–2009	consecutive	MICU	55	14	NR	NR	NR	NR	NR	NR	NR	4
17	Valle (2014)	(Valle et al., 2014) [40]	USA	2011–2013	NR	ICU	148	115	47.0	NR	Both *	NR	3	8	27.0	4
18	Okumus (2015)	(Okumus et al., 2015) [41]	Turkey	2009–2010	consecutive	ICU	74	40	55.8	NR	Both *	NR	19	NR	NR	4
19	Obi (2015)	(Obi et al., 2015) [42]	USA	2007–2012	consecutive	SICU	4844	NR	NR	NR	Pharmacological prophylaxis	NR	1947	0	NR	5
20	Mucino-Bermejo (2015)	(Muciño-Bermejo et al., 2015) [43]	Mexico	2007–2012	NR	ICU	66	33	61.1	NR	NR	NR	5	NR	NR	3
21	Allen (2016)	(Allen et al., 2016) [44]	USA	2011–2014	NR	TICU	402	300	47.0	NR	Both *	NR	12	14	28.0	3
22	Akaraborworn (2016)	(Akaraborworn et al., 2016) [6]	Thailand	2011–2013	NR	SICU	4652	2728	NR	NR	Both *	NR	NR	NR	NR	4
23	Hamada (2017)	(Hamada et al., 2017) [45]	France	2015–2016	consecutive	SICU	153	110	44.6	26.1	Both *	34.9	6	22	23.7	4
24	Karcutskies (2017)	(Karcutskies et al., 2017) [46]	USA	2011–2015	NR	ICU	1137	NR	43.6	NR	Both *	NR	17	28	20.9	5
25	Karcutskie (2017)	(Karcutskie et al., 2017) [47]	USA	2015–2017	NR	ICU	194	156	38.7	NR	Pharmacological prophylaxis	NR	11	23	23.0	5
26	Yumoto (2017)	(Yumoto et al., 2017) [48]	Japan	2013–2016	consecutive	EICU	204	144	NR	NR	Pharmacological prophylaxis	NR	NR	12	NR	4
27	Fu (2018)	(Fu et al., 2018) [49]	China	2016–2017	NR	ICU	151	100	NR	NR	NR	NR	20	NR	NR	5
28	Wilsey (2019)	(Wilsey et al., 2019) [50]	USA	2013–2017	NR	CICU	1085	NR	NR	NR	Pharmacological prophylaxis	NR	NR	NR	NR	4
29	Brendan Cloney (2019)	(Brendan Cloney et al., 2019) [51]	USA	2009–2015	consecutive	ICU	1269	645	58.9	NR	Both *	NR	NR	NR	NR	4
30	Beitland (2019)	(Beitland et al., 2019) [52]	Norway	2012–2016	consecutive	ICU	70	55	62.0	NR	Both *	40.4	11	NR	NR	5
31	Harahsheh (2019)	(Harahsheh et al., 2019) [53]	Australia	2015–2017	NR	ICU	215	131	NR	NR	Both *	NR	NR	NR	NR	4
32	Tachino (2019)	(Tachino et al., 2019) [54]	Japan	2013–2016	consecutive	ICU	859	547	58.0	NR	Both *	NR	42	106	NR	6
33	Brandi (2020)	(Brandi et al., 2020) [55]	Switzerland	2012–2015	consecutive	SICU	177	130	50.1	NR	physical prophylaxis	46.0	NR	NR	23.5	4
34	Gibson (2020)	(Gibson et al., 2020) [56]	USA	2014–2015	consecutive	MICU	243	140	64.3	29.8	Both *	NR	37	NR	NR	3
35	Dhillon (2021)	(Dhillon et al., 2021) [57]	USA	2013–2015	NR	ICU	131	92	55.6	NR	Pharmacological prophylaxis	NR	NR	NR	30.6	5
36	Chu (2021)	(Chu et al., 2021) [58]	USA	2001–2012	NR	ICU	848	474	NR	NR	NR	NR	123	NR	NR	5
37	Ow (2021)	(Ow et al., 2021) [59]	Britain	2009–2016	consecutive	LICU	623	383	NR	NR	Pharmacological prophylaxis	NR	77	NR	NR	4
38	Phan (2022)	(Phan et al., 2022) [60]	USA	2014–2018	NR	NICU	266	106	57.0	30.0	NR	NR	NR	NR	NR	4
39	Mei (2022)	(Mei et al., 2022) [61]	China	2011–2022	NR	ICU	560	365	NR	23.9	Both *	NR	NR	NR	NR	4
40	Dibiasi (2022)	(Dibiasi et al., 2022) [62]	Austria	2015–2018	NR	ICU	1352	838	NR	NR	NR	NR	NR	NR	NR	4
41	Barton (2022)	(Barton et al., 2022) [63]	USA	2017–2019	NR	ICU	91	55	NR	NR	Pharmacological prophylaxis	NR	NR	NR	NR	5
42	Abdel (2022)	(Abdel et al., 2022) [64]	Saudi Arabia	2016–2019	NR	ICU	322	306	31.3	23.9	Pharmacological prophylaxis	NR	NR	NR	28.9	5

CICU = cardiovascular intensive care unit, MICU = medical intensive care unit, SICU = surgical intensive care unit, TICU = trauma intensive care unit, NICU = neuroscience intensive care unit, LICU = liver intensive care unit, EICU = emergency intensive care unit, ISS = injury severity score, SAPS II = simplified acute physiology score. NR: not reported. BMI: body mass index. * Simultaneous use of mechanical and pharmacological prophylaxis.

**Table 2 jcm-11-06691-t002:** Subgroup analyses of prevalence of VTE in ICU patients.

Subgroups	Categories (No. of Studies)	Prevalence (%)	95% CI (%)	Sample Size	Events	*I*^2^ (%)	*p* Value within Subgroup	Q (*p* Value across Subgroups)
Year of survey	1971–2013 (22)	9.1	5.3–15.2	17,394	1122	98.34	<0.001	0.472 (0.492)
	2014–2022 (20)	11.6	7.2–18.4	9950	1096	98.25	<0.001	
Country	Asia region (8)	9.5	2.8–27.7	7065	534	99.14	0.001	1.958 (0.58)
	European (7)	14.4	6.9–27.6	1596	192	95.95	<0.001	
	North America (24)	10.3	7.0–15.0	17,080	1451	95.95	<0.001	
	Oceania (3)	5.0	1.2–18.2	1603	41	97.83	<0.001	
Obesity (%)	>26.8 (4)	17.8	8.3–34.0	5341	475	97.40	0.001	1.41 (0.24)
	≤26.8 (5)	9.7	4.8–18.6	2745	237	96.24	<0.001	
Abnormal coagulation (%)	>39.18 (4)	16.5	9.4–27.3	1793	216	94.15	<0.001	1.00 (0.32)
	≤39.18 (4)	9.2	3.1–24.0	2077	166	97.65	<0.001	
Lower extremity fracture (%)	>23.04 (4)	22.4	13.7–34.4	809	159	91.05	<0.001	4.02 (0.05)
	≤23.04 (5)	9.0	4.0–18.9	2805	239	97.09	<0.001	
Pelvic fracture (%)	>20.10 (4)	16.0	8.8–27.3	850	126	91.38	<0.001	0.28 (0.60)
	≤20.10 (5)	12.2	5.2–26.0	2764	272	97.82	<0.001	
Operations (%)	>28.74 (4)	23.9	15.5–34.9	646	148	86.89	<0.001	1.66 (0.20)
	≤28.74 (4)	14.2	6.9–27.1	580	112	90.19	<0.001	
CVC (%)	>54.29 (4)	19.7	11.0–32.8	458	114	85.31	<0.001	0.23 (0.63)
	≤54.29 (5)	16.3	9.3–27.1	5665	502	95.96	<0.001	
ICU LOS	<8 (4)	7.6	5.7–10.1	6502	514	79.68	<0.001	0.06 (0.81)
	≥8 (5)	6.9	3.1–14.6	4486	185	96.18	<0.001	
Thromboprophylaxis strategy	Both * (20)	13.4	7.6–22.7	12,583	1311	98.8	<0.001	32.21 (<0.001)
	Pharmacological prophylaxis (12)	9.3	6.0–14.1	3401	267	91.34	<0.001	
	Mechanical prophylaxis (1)	31.9	25.8–38.6	204	65	0.00	<0.001	
SAPS	>40.4 (1)	13.0	8.8–18.8	177	23	0.00	<0.001	15.01 (<0.001)
	≤40.4 (2)	29.6	24.0–36.0	223	66	0.00	<0.001	

Q = Cochran’s Q. Bold values = *p* < 0.05; CVC = central venous catheter, ICU LOS = ICU length of stay, SAPS II = simplified acute physiology score. * Simultaneous use of mechanical and pharmacological prophylaxis.

## Data Availability

The data that support the findings of this study are available from the corresponding author upon reasonable request.

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
