# Peer review of "Prevalence of Venous Thromboembolism in Intensive Care Units: A Meta-Analysis"

_jcm, 2022, doi:10.3390/jcm11226691_

Round 1

Reviewer 1 Report

The authors conducted an exhaustive review of 42 studies about the topic.

They evaluated the results by an accurate statistical analysis and discussed the evidences honestly , giving plausible explanations when results were different from what expected.

The limitations of this review are clear : some publications with small sample size, retrospective studies, different strategies of thromboprophylaxis and different populations of patients; but this meta-analysis can contribute to highlight an important unmet clinical need in ICUs: the prevention of thromboembolism in critical patients.

Author Response

We thank the Reviewers for the helpful comments on our manuscript. We have revised the manuscript according to the comments and suggestions. In the revised manuscript, all the changes are marked in yellow color for easy inspection.

Reviewer 2 Report

The idea of the article is interesting from a statistical point of view, but  few scientific details are provided.

I mention certain errors of expression from a medical point of view, as well as a number of phrases that are prone to erroneous interpretations.

I have inserted some observations  directly into the text.

The discussions are too brief and limited.

Conclusions are too general and already known,

The article has more statistical than scientific value

Author Response

e thank the Reviewers for the helpful comments on our manuscript. We have revised the manuscript according to the comments and suggestions. In the revised manuscript, all the changes are marked in yellow color for easy inspection.

Reply to Reviewer #1

I mention certain errors of expression from a medical point of view, as well as a number of phrases that are prone to erroneous interpretations.

Authors’ reply: The errors of expression mentioned were devised as advice.

I have inserted some observations  directly into the text.

Authors’ reply: The inserted observations were devised as advice.

The discussions are too brief and limited.

Authors’ reply: The discussions were devised as advice.

Conclusions are too general and already known,

Authors’ reply: Conclusions were devised to ” In conclusion, VTE was common in intensive care unit worldwide and moderated by certain demographic and clinical factors, including age, sex, thromboprophylaxis strategy, SAPS II, malignancy, spinal cord injury, and ISS. Considering the negative impacts of VTE on diagnosis, treatment, morality and prognosis, we need to pay more attention to high-risk populations of VTE. Ways to preventing VTE are pharmacological and/or mechanical now, and they require the right time, dosage and duration and also need to be accompanied by good clinical care. However, more effectively, detailed and individualize measures, such as early assessment scale and specialized rehabilitation, should be taken to reduce VTE in intensive care unit”.

Round 2

Reviewer 2 Report

I have read the new version of the article

I have inserted some of my comments into the text

Some phrases must be reformulated

Author Response

We thank the Reviewers for the helpful comments on our manuscript. We have revised the manuscript according to the comments and suggestions. In the revised manuscript, all the changes are marked in green color for easy inspection.